



# Altitudinal variation in impacts of snow cover, reservoirs and precipitation seasonality on monthly runoff in Tibetan Plateau catchments

Nan Wu[a,b,c,f], Ke Zhang[a,b,c,d,e,*], Amir Naghibi[f], Hossein Hashemi[f], Zhongrui Ning[b,c], Jerker Jarsjö[g]

[a]*The National Key Laboratory of Water Disaster Prevention, Hohai University, Nanjing, Jiangsu,*

*210024, China*

[b]*Yangtze Institute for Conservation and Development, Hohai University, Nanjing, Jiangsu, 210024,*
*China*

[c]*College of Hydrology and Water Resources, Hohai University, Nanjing, Jiangsu, 210024, China*

[d]*China Meteorological Administration Hydro-Meteorology Key Laboratory, Hohai University, Nanjing,*

*Jiangsu, 210024, China*

[e]*Key Laboratory of Water Big Data Technology of Ministry of Water Resources, Hohai University,*
*Nanjing, Jiangsu, 210024, China*

[f]*Division of Water Resources Engineering, LTH, Lund University, Lund, 22100, Sweden*

[g]*Department of Physical Geography, Stockholm University, Stockholm, 10691, Sweden;*

* Corresponding author at: The National Key Laboratory of Water Disaster Prevention, Hohai
University, Nanjing, Jiangsu, 210098, China

E-mail address: kzhang@hhu.edu.cn (Ke Zhang)





## ABSTRACT

Although of great importance for long-term, effective water resource allocation, current

knowledge of monthly runoff variability, its spatio-temporal characteristics, and underlying

key drivers, including their sensitivity to climate change and other human impacts, is limited.

With a particular focus on 10 sub-basins along an elevation gradient (1000 to 5900 m.a.s.l.) in

the hydrologically complex, seasonally cold Yalong River basin, China, this study developed

an extended Budyko framework based on monthly water balances (2002-2016) to consider

snow storage dynamics ($\Delta S_{snow}$) separately from other terrestrial water storage changes ($\Delta S'$),

including those related to hydropower reservoir construction. Results showed that snow

accumulation and snowmelt are main drivers of runoff seasonality in the upper sub-catchments

of the Yalong River basin, with propagating impacts also on lower-elevation snow-free sub-

catchments, which are increasingly under the additional influence of hydropower reservoirs.

This creates a relatively strong altitudinal heterogeneity in drivers of monthly runoff, which

has been hypothesized to occur also in other world regions including e.g. major European rivers

of Alpine origin, although not yet quantified at similarly high spatio-temporal resolution.

Furthermore, an observed decrease in runoff seasonality in the Yalong River at its Yangtze

River outlet (that receives water from all 10 investigated sub-basins) was shown to be unrelated

to snow storage changes and hence likely caused by trends in unfrozen precipitation seasonality

and/or flow-modulating impacts of constructed reservoirs, natural lakes and groundwater,

implying that further snow thinning may exacerbate such trends in the future. Implementing

the variance decomposition method based on the extended Budyko framework, the intra-annual



runoff variability ($\sigma^2_R$) was captured by calculating the variance and covariance of influencing

factors ($R^2$ values above 0.9 in most sub-basins) with the main contributors being variances of

rainfall *($P_r$)* and $\Delta S'$. Methodologically, we have verified the substantial contribution of

hydropower reservoir storage changes on total storage changes by independent analysis of

reservoir storage data, supporting the applicability of the extended monthly Budyko framework

for identifying dominant processes in the context of runoff generation and the rapid

environmental changes that the Yalong River basin and other cold regions (not least of the

Tibetan plateau) are currently experiencing.

**Keywords:** Runoff variability, Snowmelt, Terrestrial storage change, Altitudinal trend,

Correlation analysis, Budyko framework, Yalong River basin

## 1. Introduction

Runoff is a key component of the hydrological cycle and is highly susceptible to external

environmental factors, primarily climate change and human activities, which can lead to

significant changes in hydrological processes (Bao et al., 2023). Climate variations such as

precipitation intensity, rising temperature, and enhanced radiation not only affect the water

vapor content in the atmosphere but also alter the surface characteristics of catchments (Li and

Quiring, 2021). Moreover, extensive human activities such as reservoir construction increase

the complexity of surface water flow, making accurate analysis of the hydrological cycle a

challenge (Gutenson et al., 2020). Observations indicate that 24% of the global river flow has

undergone significant changes (Li et al., 2020). Runoff changes, influenced by evolving factors

(Yao et al., 2020), show the sensitivity of hydrological cycles to climate conditions and surface



characteristics (Huang et al., 2021). While intra-annual analysis of runoff response to climate

variability and change provides vital information for the effective allocation of water resources,

a significant amount of research is focused on multi-annual timescale analysis (Huo et al., 2021;

Ning et al., 2017; Tu et al., 2015). Changes in monthly climate conditions and human activities

largely drive variations in monthly runoff (Xin et al., 2019; Yao et al., 2020).

In many parts of the world, seasonal storage of snow and ice provides meltwater and

secures water supply over the growing season. Although climate change impacts frequently act

to reduce seasonal volumes of stored snow and ice, it is challenging to predict the consequences

for runoff. This is partly due to the difficulty of disentangling such impacts from other

influences on runoff, such as changes in precipitation and temperature, and reservoir

construction. Benchmark estimates have shown that that snowmelt during the rainy season

(April to June) contributed to 31% of the annual runoff for the source regions of the Indus River,

while the source regions of the Yellow River, Yangtze River, Mekong River, Thanlwin River,

and Brahmaputra River received snowmelt from April to June, contributing to 20-23% of the

annual runoff (Zhang et al., 2013). Additionally, the snowfall-to-precipitation ratio has been

found to exert a significant influence on both annual runoff and intra-annual runoff variation

(Berghuijs et al., 2014; Liu et al., 2022; Zhang et al., 2015).

Recent studies on this topic have also shown evidence of decreased runoff seasonality, e.g.

in snow-dominated rivers of central Europe (Rottler et al., 2020). In cold regions of China,

water storage and runoff characteristics show quite complex spatio-temporal patterns over the

last 30 years (Fang et al., 2019). This is particularly the case in the downstream regions that

use reservoirs to alter the intra-annual hydrological cycle and store water to ensure a sustainable



water supply during dry seasons in the face of agricultural, industrial, and domestic demands (Shen, 2018). More generally, water storage in lakes, reservoirs and groundwater aquifers may contribute considerably to monthly runoff dynamics (Bai et al., 2018; Hwang and Devineni, 2022; Shi and Gao, 2022). It is hence most likely that distinct differences in drivers of monthly runoff exist across altitudinal gradients of different mountainous regions of the world (Kuhn et al., 2016; Rottler et al., 2020; Shen, 2018), although the details of such patterns are largely unexplored. Zhang et al. (Zhang et al., 2016) emphasized that main factors (e.g., human activities and climate-driven changes in runoff) that affect the runoff variance deserve more attention. A more detailed understanding intra-annual runoff characteristics in mountainous watersheds under changing environments combined with attribution analyses are crucial for sustainable water resource management (Dethier et al., 2020; Liu et al., 2017).

Precipitation and potential evapotranspiration define catchment water availability and storage capacity (Huang et al., 2021; Li et al., 2021; Yao et al., 2020). On a multi-year scale, precipitation is partitioned into evapotranspiration and runoff, reflecting competition between water supply (precipitation) and available energy (potential evapotranspiration) and regulated by the corresponding underlying surface characteristics (Wu et al., 2018). To analyze annual and multi-year scale hydrological processes considering rainfall and runoff changes, the Budyko framework has been widely used (Choudhury, 1999) The framework was initially developed to address water resource constraints through multi-year averages (Kazemi et al., 2021; Wang and Tang, 2014; Yang et al., 2008; Zhang et al., 2004) and has been widely applied to quantitatively analyze the impact of climate change and human activity on runoff (Liu et al., 2021; Wang and Tang, 2014; Xu et al., 2014). However, since this framework assumes multi-



year, steady-state conditions, it is not applicable at intra-annual timescales. Furthermore, most

assessments investigate the main drivers only by comparing runoff sensitivity, rather than their

relative contribution to the actual variance of runoff (Liu et al., 2019).

Recently, some studies have attempted to extend the Budyko framework to intra-annual

timescales, mainly by including monthly terrestrial water storage changes as part of the water

supply (Du et al., 2016; Liu et al., 2019) and establishing a new water supply-demand

relationship on monthly scale (Huang et al., 2021; Wu et al., 2019). Other studies have also

integrated monthly terrestrial water storage in water balance using the Budyko framework

along with hydrological models (Yao et al., 2020; Zhang et al., 2020). Some studies have

demonstrated that incorporating terrestrial water storage significantly enhances predictability

compared to the previous precipitation and potential evapotranspiration relationship (Wu et al.,

2018; Zhang et al., 2010). Although the extended Budyko framework has been effective to

analyze monthly runoff changes, the attribution analysis of intra-annual runoff changes needs

to consider more driving factors, particularly due to the complexity and interaction of climate

change and human activities in the cold mountainous regions (Liu et al., 2018; Luo and Lau,

2018). Equally important from a process understanding perspective is the consideration of

sufficiently fine-resolved temporal and spatial scales (Fang et al., 2016).

The Qinghai-Tibet Plateau, often referred to as the "Third Pole," is a region where the

atmosphere, hydrosphere, cryosphere, and biosphere intricately interact (Bao et al., 2023; Cui

et al., 2023). Snow accumulation begins in the fall, lasting until the subsequent spring, and in

some high-altitude areas, snow cover persists even into the summer months (Wang et al., 2018;

Wu et al., 2012). Snowmelt runoff stands as a crucial component of the primary runoff source





on the Qinghai-Tibet Plateau, exhibiting distinct seasonal variations and primarily impacting

spring runoff (Gao et al., 2017; Han et al., 2019).

We here consider the Yalong River basin, situated between 1000 and 5900 m.a.s.l. in the

southeastern part of the Tibetan Plateau, which experiences seasonal snow cover in its upstream

regions, while the construction of downstream cascade hydropower stations has greatly

affected terrestrial water storage capacity. These factors exert significant influences on regional

water cycling, especially at the intra-annual scales (Wang et al., 2018; Wu and Shen, 2007).

Therefore, this study aims to (1) extend the Budyko framework to distinguish between snow

storage change ($\Delta S_{snow}$) and remaining water storage change ($\Delta S'$) in monthly water balances;

(2) determine   interannual and intra-annual variations of hydrological variables including

runoff ($R$), rainfall ($P_r$), snowmelt ($S_{melt}$), evapotranspiration ($E$) and terrestrial water storage

change ($\Delta S$) in nested catchments along a pronounced altitudinal gradient, and (3) examine

how relationships between runoff and other factors can vary with elevation, including

assessments of major contributors to runoff variation (using variance decomposition analysis);

and (4) discuss wider implications including the susceptibility of the identified monthly runoff

contributors to climate-change and other human impacts.

## 2. Methods and application

In this section, the theoretical framework for attributing runoff variability based on the

extended Budyko is described in Sect. 2.1. Then, the implementation of the extended Budyko

framework is explained in Sect. 2.2. Table 1 presents the variables and acronyms used in this

study.





**Table 1.** Description of hydrological variables and acronyms.

| Number | Variable (mm) | Abbreviation |
|:---:|:---:|:---:|
| 1 | Precipitation | $P$ |
| 2 | Rainfall component of $P$ | $P_r$ |
| 3 | Snowfall component of $P$ | $P_s$ |
| 4 | Total available water | $P'$ |
| 5 | Potential evapotranspiration | $E_0$ |
| 6 | Actual evapotranspiration | $E$ |
| 7 | Runoff | $R$ |
| 8 | Change in terrestrial water storage | $\Delta S$ |
| 9 | Change in ground snowpack storage | $\Delta S_{snow}$ |
| 10 | Differences between $\Delta S$ and $\Delta S_{snow}$ | $\Delta S' = \Delta S - \Delta S_{snow}$ |
| 11 | Snow melt | $S_{melt}$ |

## 2.1. Methods

Figure 1 provides an overview of the extended Budyko calculation for the present study.

Observation data on temperature $T$, precipitation $P$ and runoff $R$ were used to resolve monthly

water balances, and for estimation of multiple hydrological variables (included in Table 1) in

ten sub-basins of the Yalong River basin. In addition to two relatively large headwater

catchments and four smaller ones, we here consider four nested catchment areas (see further

the study area description of Sect. 2.2.1) for which the net runoff $R$ at any particular time is

quantified as the difference in measured discharges $\Delta Q = Q_{down} - Q_{up}$ between the downstream

discharge station and upstream discharge station, divided by the corresponding difference in

catchment areas $\Delta A = A_{down} - A_{up}$. For the headwater catchments, $Q_{up}$ and $A_{up}$ are per definition

equal to zero. The water balance variables were then used as input to an extended Budyko

framework. Subsequently, the correlation between monthly $R$ and the hydrological variables

(including drivers of $R$, such as $P_r$, $S_{melt}$, $\Delta S$, and $E$; Table 1) was explored, focusing on the lag

time of $R$ response to $P$ events. Finally, the relative contributions of driving factors to monthly





runoff variability were quantitatively assessed using the variance decomposition method. The

hydrological variables employed in this study are shown in Table 1.

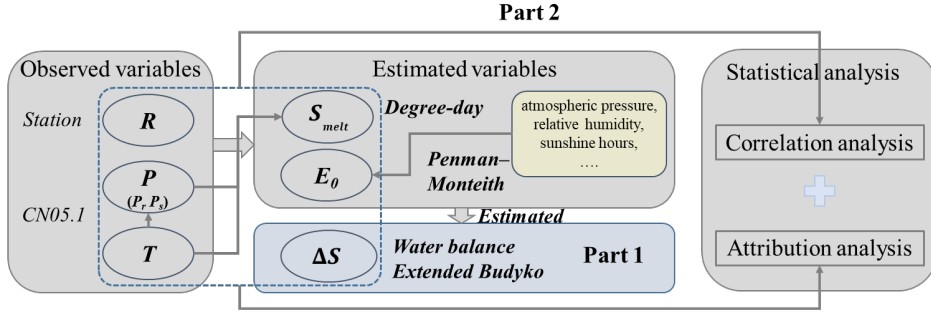

**Figure 1.** Flowchart of the extended Budyko framework. Italic text represents the data source, and italic-

bold text represents the calculation method for obtaining the data.

## 2.1.1 Extended Budyko framework by decomposing ΔS

The present analyses use water balance methods based on the Budyko hydrothermal

coupling theory, which compared with statistical empirical methods possesses substantial

physical significance and is straightforward to calculate and parameterize (Hwang and

Devineni, 2022; Shi and Gao, 2022). The partitioning of annual precipitation into annual

evapotranspiration and streamflow is determined by the competition between available water

and available energy measured by potential evapotranspiration (Huang et al., 2021), and many

scholars have proposed empirical equations to characterize this relationship that is a part of the

Budyko framework (Choudhury, 1999; Wang and Tang, 2014; Yang et al., 2008; Zhang et al.,

2004), However, due to their simplicity and effectiveness, some of those equations have been

applied more than others. We consider one of the most popular equations, namely Choudhury-

Yang:



$$E = \frac{P \times E_0}{(P^n + E_0{}^n)^{1/n}} \tag{1}$$

where potential evapotranspiration ($E_0$, mm) and precipitation ($P$, mm) act as indicators

for energy and water supply, respectively, and where $E$ (mm) is the calculated actual

evapotranspiration, and $n$ is a landscape parameter, mainly expressing impacts of prevailing

surface conditions within the basin.

The water balance equation on the decadal time scale for a basin can be expressed using

equation (2), wherein $P$ represents the total water input to the basin. The outputs comprise

runoff ($R$, mm) and evapotranspiration ($E$, mm).

$$P = R + E \tag{2}$$

The change in terrestrial water storage can be assumed to be approximately 0 on the

decadal time scale, while on the intra-annual time scale, it cannot be neglected (Huang et al.,

2021; Xu et al., 2012):

$$P = R + E + \Delta S \tag{3}$$

In order to consider the dynamic changes of monthly snow accumulation and melting

separately, $P$ was differentiated into rainfall ($P_r$, mm) and snowfall ($P_s$, mm) using daily

temperature thresholds (Widen-Nilsson et al., 2007), and the terrestrial water storage change

was also divided into two components: one accounts for snow storage change ($\Delta S_{snow}$, mm),

and the other represents the remaining storage changes ($\Delta S'$ mm; mainly including storage

changes in surface water, groundwater and soil water). Thus, Equation (3) can be expressed as:

$$P_r + P_s = R + E + \Delta S_{snow} + \Delta S' \tag{4}$$

In this study, the cumulative value of daily $P_r$ (and $P_s$) was calculated based on daily

temperature to derive monthly $P_r$ (and $P_s$). $\Delta S_{snow}$ is mainly supplied in solid form by $P_s$ and





depleted in liquid form by melting snow ($S_{melt}$, mm). Equation (4) can then be expressed as:

$$P_r + P_s = R + E + (P_s - S_{melt}) + \Delta S' \tag{5}$$

The monthly water balance equation can be written as:

$$P_r + S_{melt} = R + E + \Delta S' \tag{6}$$

Therefore, the total available water ($P'$) for $E$ and $R$ is expressed as the sum of $P_r$, $S_{melt}$

and $\Delta S'$ (Zeng and Cai, 2015). Accordingly, Equation (6) can be modified as follows:

$$P' = P_r + S_{melt} - \Delta S' = R + E \tag{7}$$

Combining equation (1) and equation (7), an extended Budyko framework with a

parameter $n$ can be formulated as:

$$E = \frac{(P_r + S_{melt} - \Delta S') \times E_0}{((P_r + S_{melt} - \Delta S')^n + E_0{}^n)^{1/n}} \tag{8}$$

In this study, the Penman-Monteith method (Allen et al., 1998) was used for $E_0$ calculation,

e.g. in consistency with the FAO Irrigation and Drainage Paper No. 56 (Allen et al., 1998)

showing that the Penman-Monteith model is a preferable physical approach to estimate $E_0$. It

is often utilized as a standard method for verifying other empirical methods (Chen et al., 2005).

The Penman-Monteith method has also been widely employed for estimating

evapotranspiration on snow surfaces (Stigter et al., 2018; Xin et al., 2021), considering it a

saturated or unstressed surface similar to water surfaces (Yang and Bai, 2023) . Due to the high

albedo of snow surface, the surface radiance is typically higher than that of the air, resulting in

negative net radiation for many snow-covered pixels. Currently, the evaporation mechanism

under negative net radiation conditions is under-studied. Therefore, in this study, $E$ is assumed

to be 0 when the net radiation is less than 0 (Gan et al., 2022).

Therefore, the monthly runoff $R$ can be expressed as:



$$R = P_r + S_{melt} - \Delta S' - \frac{(P_r + S_{melt} - \Delta S') \times E_0}{((P_r + S_{melt} - \Delta S')^n + E_0{}^n)^{1/n}} \quad (9)$$

in which $R$ was obtained from direct discharge measurements, and $S_{melt}$ was computed

using the degree-day method. The $n$ parameter is primarily employed to characterize the

underlying surface conditions of the basin, including factors such as average slope (Zhang et

al., 2004) and vegetation type or land use (Bounoua et al., 2004), which might undergo

significant changes on long-term scales, whereas differences over short-term scales, such as

monthly, are typically minimal. Therefore, based on a previous study (Huang et al. 2021) we

assumed that $n$ can be treated as constant value for monthly and multi-year scales. We

hypothesized and tested our assumption, $n$, by calculating the evapotranspiration using the

Budyko and extended Budyko. As shown in Table S1, the results showed a high correlation

coefficient between the evapotranspiration values in both models, which supports our

hypothesis. Thus, the monthly $\Delta S'$ was calculated by closing the water balance.

    The degree-day method relies on readily available data and straightforward calculations,

providing comparable accuracy to the energy balance method at the basin scale (Hock, 2003).

The degree-day model operates on the assumption of a strong positive linear correlation

between temperature and $S_{melt}$, with the fundamental equation depicted in equation (10). If the

daily temperature $T_i$ surpasses the temperature threshold $T_0$, the $S_{melt}$ is determined by the

degree-day factor ($D$, mm/°C/day) and temperature:

$$S_{melt} = D \times (T_i - T_0) \qquad \text{if } T_i > T_0 \quad (10)$$

    In this method, the parameter $D$ typically falls within the range of 2.5-14 mm/°C/day, with

specific values described in Sect. 2.2.2. $T_0$ is commonly assumed to be 0°C, and $S_{melt}$ is





constrained by the presence of existing snow depth.

### 2.1.2. Cross correlation analysis

Runoff is influenced by various natural and geographical factors, such as climate and the underlying surface. Precipitation serves as the primary driving factor in the formation of runoff. Following intricate surface and subsurface hydrological processes, catchment responses to precipitation gradually undergoes smoothing and lagging (Brutsaert and Hiyama, 2012) the hysteresis relationship between monthly precipitation and runoff within Yalong River basin, the variable $\tau$ to compute the lag time between the precipitation events and subsequent runoff changes, ranging from 0 to 5 months. Statistical significance was considered as p≤0.05 for this analysis.

### 2.1.3. Quantifying the contributions of different factors to runoff variability

Equation (9) could be expressed as $R = f(P_r, S_{melt}, \Delta S', E_0)$. The variance of $R$ (within a year is determined by the variance of each driving factor ($P_r, S_{melt}, \Delta S', E_0$) and their covariance (Ye et al., 2015):

$$
\begin{aligned}
\sigma_R{}^2 = {} & (\frac{\partial f}{\partial P_r})^2 \sigma_{P_r}{}^2 + (\frac{\partial f}{\partial S_{melt}})^2 \sigma_{S_{melt}}{}^2 + (\frac{\partial f}{\partial \Delta S'})^2 \sigma_{\partial \Delta S'}{}^2 + (\frac{\partial f}{\partial E_0})^2 \sigma_{E_0}{}^2 + 2(\frac{\partial f}{\partial P_r} \\
& \cdot \frac{\partial f}{\partial S_{melt}}) cov(P_r, S_{melt}) + 2(\frac{\partial f}{\partial P_r} \cdot \frac{\partial f}{\partial \Delta S'}) cov(P_r, \Delta S') + 2(\frac{\partial f}{\partial P_r} \\
& \cdot \frac{\partial f}{\partial E_0}) cov(P_r, E_0) + 2(\frac{\partial f}{\partial S_{melt}} \cdot \frac{\partial f}{\partial \Delta S'}) cov(S_{melt}, \Delta S') + 2(\frac{\partial f}{\partial S_{melt}} \\
& \cdot \frac{\partial f}{\partial E_0}) cov(S_{melt}, E_0) + 2(\frac{\partial f}{\partial \Delta S'} \cdot \frac{\partial f}{\partial E_0}) cov(\Delta S', E_0)
\end{aligned} \tag{11}
$$

Hence, equations (10) are utilized to compute the contributions of the variance of each driving factor ($P_r, S_{melt}, \Delta S', E_0$) and their covariance to intra-annual runoff variability for the period 2002-2016.





## 2.2. Study area and data

### 2.2.1. Study area

The Yalong River is the largest tributary of the Jinsha River, which is made up of the upper reaches of the Yangtze River. Being a basin that encompasses a wide range of altitudes (1,000 to 5,900 m.a.s.l.) and hydro-climatic conditions (see below) while being increasingly impacted by human activities including hydropower expansion, it serves as a case study to test our method. It traverses from northwest to southeast, boasting a total length of 1,570 km and a drainage basin area of 128,000 km². Meteorological observations in the upper reaches of the basin are limited due to the complex topography (Fig. 2). The precipitation and temperature in the Yalong River basin increases from north to south, with an average annual temperature of -3 in the north to 26 °C in the south in summer and -18 in the north to 14 °C in the south in winter. The precipitation ranges from 600 to 800 mm in the upper region, 1,000 to 1,400mm in the middle region, and 900 to 1,300 mm in the lower region. June to October is the rainy season. The annual average discharge at its confluence with the Yangtze River is 1,900 $m^3$/s, with an annual discharge of nearly 60 billion $m^3$, representing 13% of the total water volume upstream of the Yangtze River (He et al., 2015).

The Yalong River is renowned for its abundant hydropower resources and the middle and lower reaches are designated to the national hydropower base, ranking third among the 13 hydropower bases in China (Wu and Shen, 2007). In the downstream reaches, the construction of five hydropower stations, including Jinping I (2013), Jinping II (2013), Guandi (2012), Ertan (1999), and Tongzilin (2015), has significantly altered the terrestrial water storage (Wu and Shen, 2007). For this study, the Yalong River basin was divided into ten sub-basins based on



the distribution of hydrological stations, as depicted in Figure 2. Station coordinates and basic

hydrological and meteorological data of the respective sub-basins are given in Table S2.

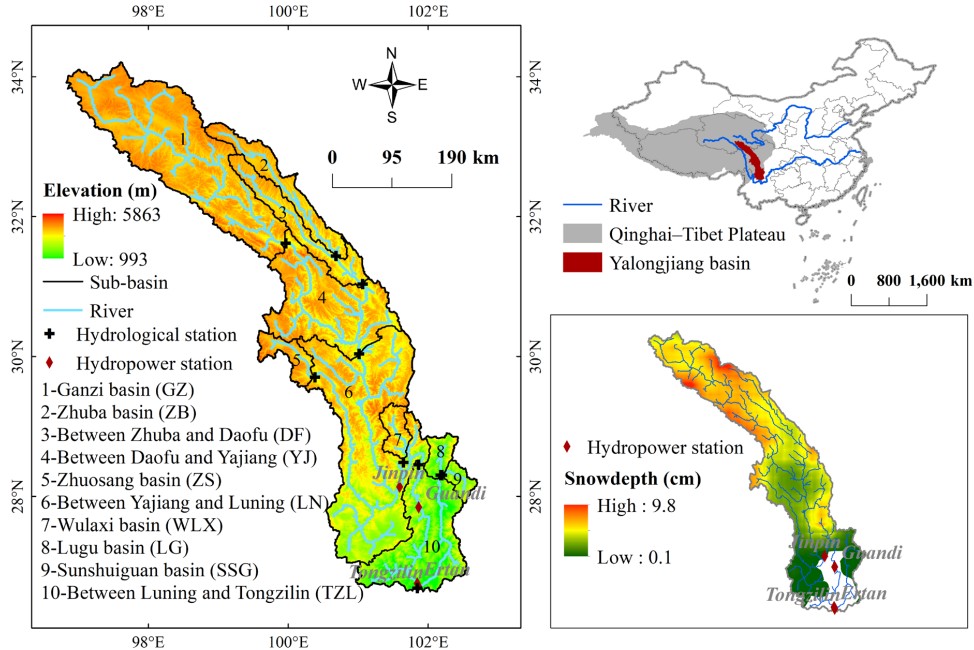


**Figure 2.** Geographic location of the sub-basins (1-10) and mean yearly average snow depth from 2002 to 2016 in the Yalong River basin.

## 2.2.2. Dataset

The average monthly runoff records during the period 2002-2016 at 10 hydrometry

stations were obtained from the China Hydrological Yearbook provided by the Ministry of

Water Resources of China. Table S1 gives information about the locations of stations and their

corresponding sub-basins, including runoff, precipitation, temperature, and potential

evapotranspiration. The first column represents the labels of the 10 sub-basins, as depicted in

Figure 2, with the Tongzilin Hydrological Station serving as the controlling station for the

entire Yalong River basin.





For meteorological data, the CN05.1 dataset (1961-2020) was utilized in this study, which provides a daily grid resolution of 0.25°×0.25°, covering various meteorological variables, including precipitation, temperature, atmospheric pressure, relative humidity, sunshine hours, and wind speed across China. The dataset has been generated by interpolation of data from

over 2,400 observation stations throughout China using the "abnormal method" (New et al., 2000). This study used a dataset of the spatial distribution of degree-day factors for glaciers and snow in High Mountain Asia, which was derived from observations over 40 glaciers. The spatial resolution of this dataset is 0.5°, with units of mm/℃/day (Zhang et al., 2019). The evapotranspiration data was derived from GLEAM (Global Land-surface Evaporation: the

Amsterdam Methodology) (1980-2020), which combines a wide range of remote sensing observations to derive daily actual evapotranspiration and its different components, including snow sublimation (Miralles et al., 2011).

## 3. Results

### 3.1. Intra-annual changes in hydrological variables

Figure 3 depicts the full time series of $P_r$, $S_{melt}$, $E$, $R$, $\Delta S_{snow}$, and $\Delta S$ of 10 sub-basins within the Yalong River basin from 2002 to 2016. It should be noted that $R$ refers to the net runoff generated locally within the sub-basin, which in case of nested basins (DF, YJ, LN and TZL) excludes flows generated by areas further upstream. Figure 3 shows that $P_r$ exhibited pronounced interannual variation and was additionally higher in downstream areas. $S_{melt}$ and

$\Delta S_{snow}$ primarily occurred in the middle and upper reaches, including upstream sub-basins such as GZ, ZB, DF and YJ (Figure 3). Lower reaches subbasins with $\Delta S_{snow}= 0$ (WLX, LG, SSG,





TZL) would hence per definition have $\Delta S = \Delta S'$. Within upstream areas $E$ approached zero in the winter. Notably, the $R$ and $\Delta S$ in WLX underwent an abrupt change in 2009, demonstrating a significant decrease in $R$ and general increases in $\Delta S$. The $R$ decrease may in part be caused

by filling of reservoirs and reservoir-induced increases in $\Delta S$. In LN, the $R$ and $\Delta S$ began to change around 2013, with locally created $R$ fluctuating gently and $\Delta S$ starting to show pronounced intra-annual fluctuations including considerably higher positive values, which is consistent with the construction period of the Jinping hydropower station including the filling of the dam and the start of intra-annual flow regulations. Additionally, the periodic

characteristics of $R$ (more runoff in summer and less in winter) in lower elevation regions of LG and TZL were not as clear as in other regions, and the amplitude of $\Delta S$ varied significantly, including sharp transitions. There were even individual negative values of net $R$ in TZL. Such phenomena are closely related to the influence of reservoirs (including hydropower dams) on the net runoff in nested catchment segments, where monthly outflow may be lower than

monthly inflow, for instance when a hydropower dam near the outlet of a nested catchment temporarily stores more water ($\Delta S_{dam}$) than what is created from local (positive) $R+\Delta S$ just upstream of the dam.

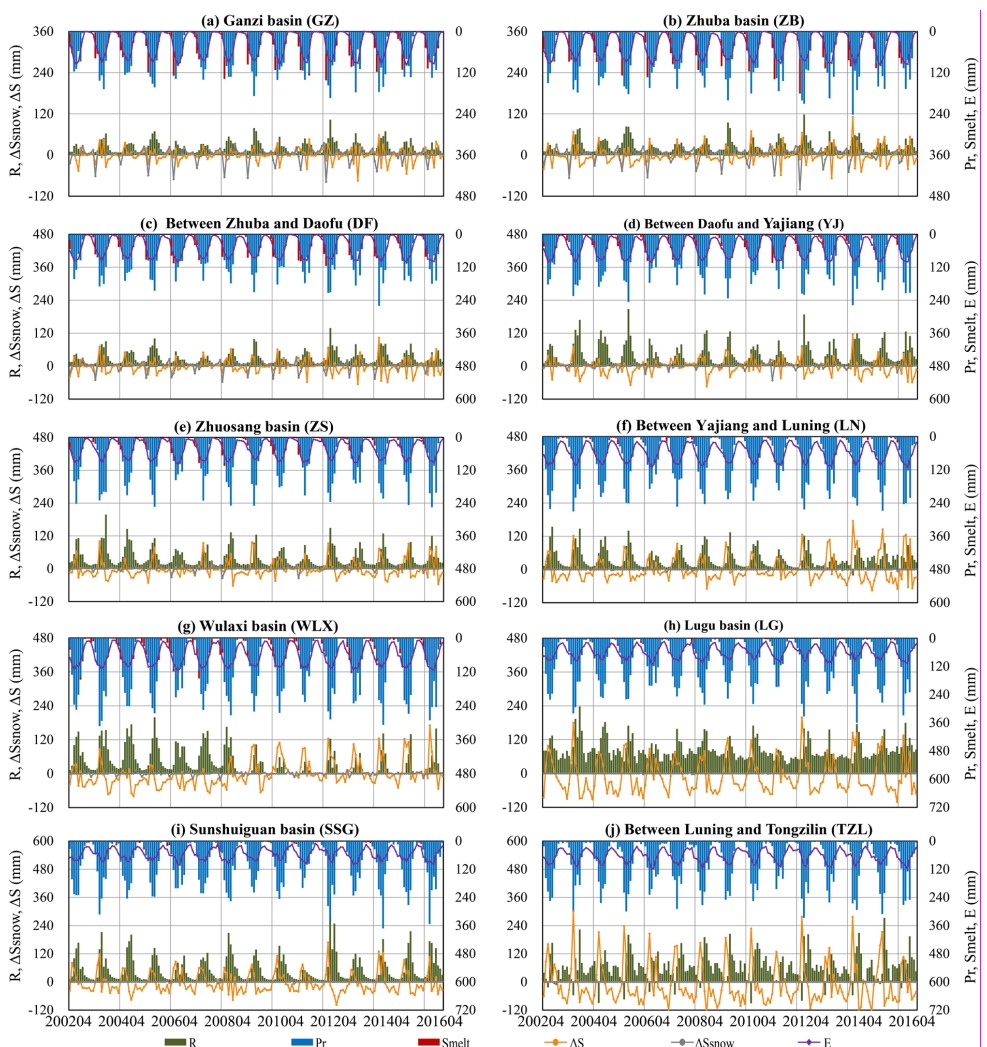

**Figure 3.** Time series of runoff ($R$), rainfall ($P_r$), snowmelt ($S_{melt}$), total storage change ($\Delta S$), snow storage

change ($\Delta S_{snow}$) and evapotranspiration ($E$) from 2002 to 2016 in the 10 sub-basins.

To further examine the intra-annual characteristics of key hydrological variables, Figure

4 illustrates the average monthly time series of $P$, $P_r$, $S_{melt}$, $\Delta S$, $E$, and $R$ across the 10 sub-

basins from 2002 to 2016. Differences between $P$ and $P_r$ are due to $Ps$ and occurred primarily

in the middle and upper regions including GZ and ZB during October to April. The $S_{melt}$, which



is mainly caused by rising spring temperatures, reflects a delayed impact of $P_s$ on runoff. Peak values of $S_{melt}$ occurred between March and May. Constrained by the potential evapotranspiration and water supply conditions, $E$ steadily increased from January to July, reaching its peak value in July before declining until December. $R$ was positively correlated to $P_r$ as as shown in Section 3.2, while $\Delta S$ primarily was positive during spring and summer,

reflecting water storage. The contrasting negative $\Delta S$ during autumn and winter reflects water release. The maximum positive value of $\Delta S$ was typically observed in June because of the $P_r$ and $S_{melt}$ characteristics. The intra-annual variability of $\Delta S$ was more pronounced in the downstream basin compared to the upstream, which emphasizes the increasing impacts of flow regulation. The net $R$ in the nested TZL-catchment exhibited a minimum (negative) value in

June, coinciding with a maximum value of $\Delta S$. This reflects a situation where the local water storage consists of considerable water volumes created upstream of the nested catchment itself. Such storage ensured high $R$ in LG and TZL during the dry season.

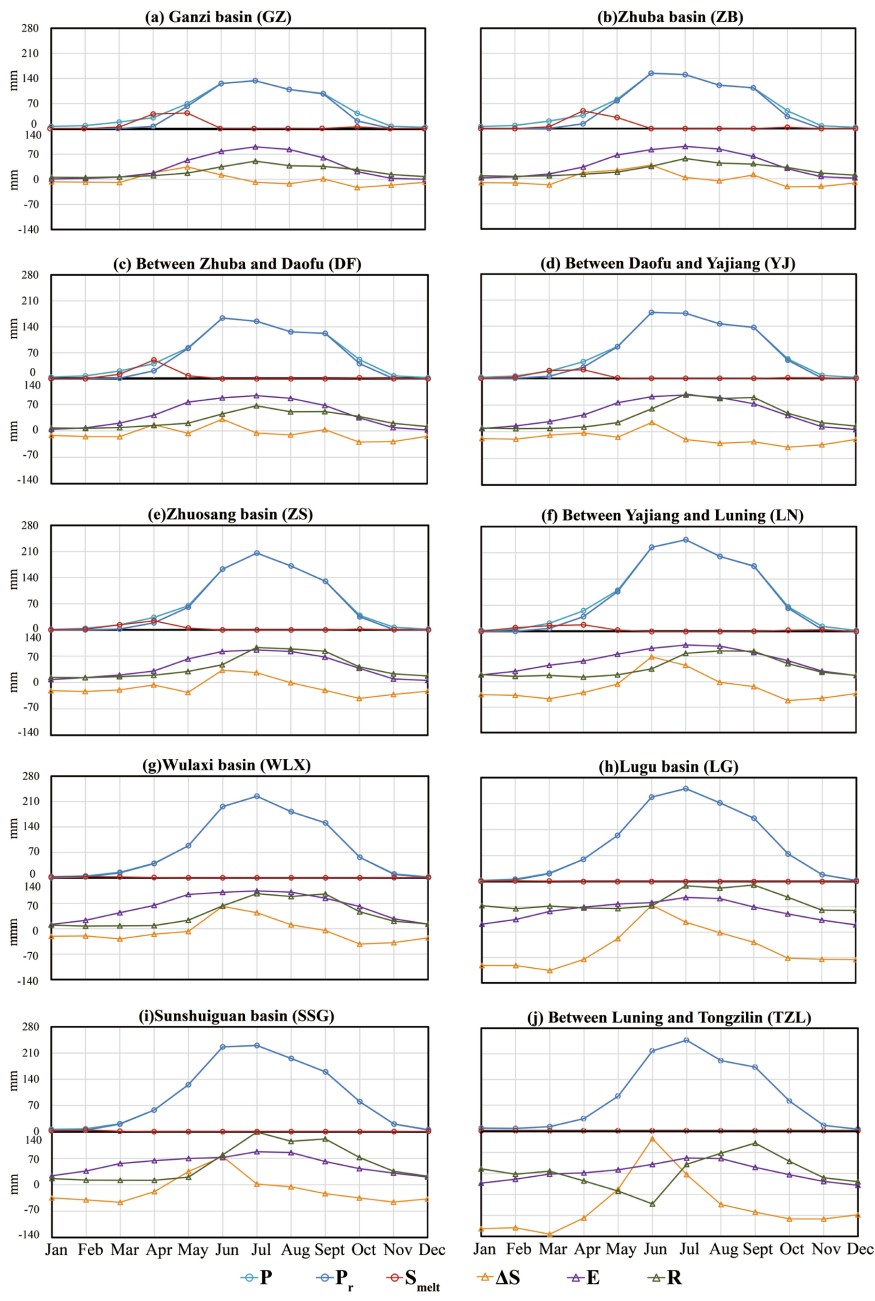

**Figure 4.** Annual average monthly $P$, $P_r$, $S_{melt}$, $\Delta S$, $E$, and $R$ from 2002 to 2016.

The relative contribution of the different water balance terms varies across seasons and





regions (Figure S1). $P_r$ was generally the main contributor across catchments during May-October, while in most catchments $\Delta S'$ was the main contributor during January-April and November-December. It is noteworthy to mention that the proportions of $P_r$ and $S_{melt}$ in the high elevation catchments of GZ (in January, February, November, and December), ZB (in January, February, and December), DF (in January and December), and YJ (in December) were all 0%, as snow accumulation due to temperatures below 0°C essentially prevented effective water input into these basins. During this period, $\Delta S'$ were primarily driven by the output terms of $R$ and $E$, although their values were low (Fig. 4).

## 3.2. Monthly runoff response to different factors

Partial correlation analysis was conducted after downtrending to examine relationships between monthly $R$ and each of its driving factors $P_r$, $S_{melt}$, $\Delta S$, and $E$. As shown in Figure 5, the results revealed strong positive correlations between $R$ and $P_r$, however mainly around the summer period (May-Sept) only. There were considerable differences between the catchments, with the highest correlation coefficient observed for the low-elevation small headwater catchment of SSG between March and September (reaching 0.87 in May) and the overall lowest correlations found in the large central and nested catchment of LN where statistical significance was only obtained for May. Conversely, robust negative correlations between $R$ and $\Delta S$ were observed from September to January. The correlations were remarkably strong throughout almost all subcatchments. They were e.g. significant in 8 out of 10 catchments in November, 10 out of 10 in December (including a peak correlation of -0.98 in TZL), and 9 out of 10 in January. The effect reflects an increased dominance of $\Delta S$ as a source of $R$, which is in wintertime can be explained by negligible $P_r$, $S_{melt}$ and $E$. Furthermore, in the downstream





catchments including LG and TZL, $\Delta S$ continues to be a dominating source of R throughout

the year, reflecting that $\Delta S$ most likely boosted by reservoir storage to values well above those

of $P_r$, $S_{melt}$ and $E$. This is hence in contrast to upstream catchments including GZ, ZB, DF, YJ,

and ZS, which with few exceptions do not exhibit significant correlations between $R$ and $\Delta S$

between March and August. From January to May, $S_{melt}$ served as an additional contributor to

locally created $R$ during one or more months in all sub-basins except for the downstram-most

TZL (Fig. 5). Regarding correlations between $R$ and $E$ they were found to be positive

particularly in spring and early summer, probably because an increased availability of effective

water supports simultaneous increases in $R$ and $E$. In August however, correlations were

negative in all of the investigated basins (Fig. 5) implying pronounced losses to the atmosphere.



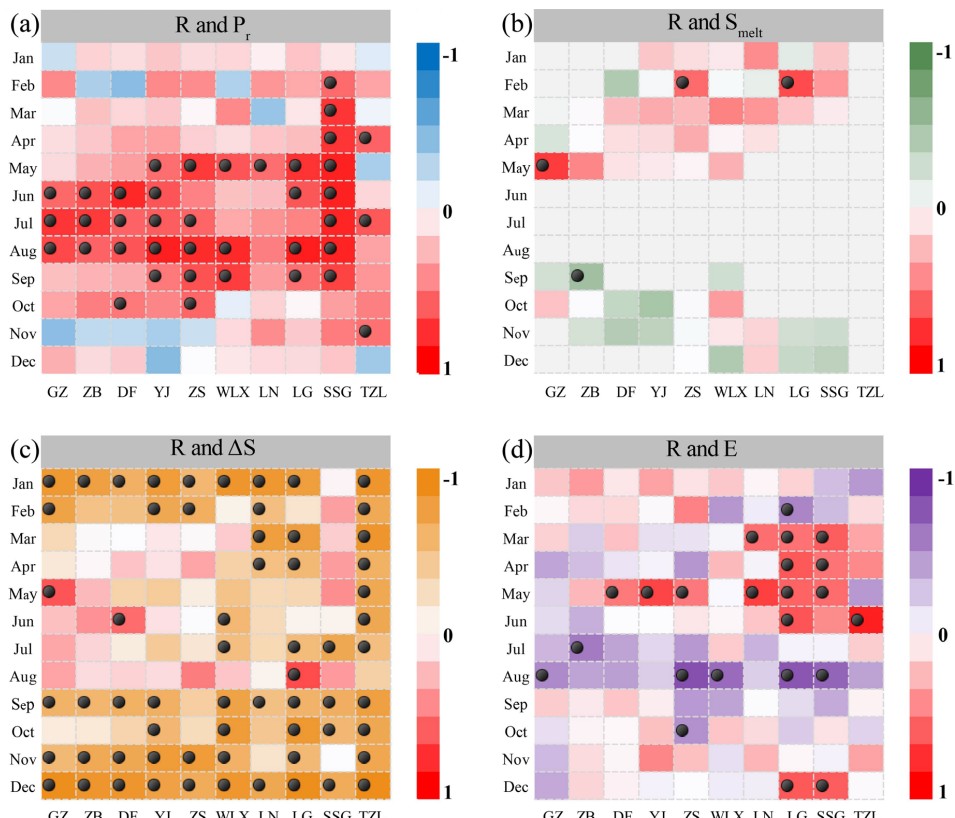

**Figure 5.** Correlation matrices between monthly $R$ and its driving factors ($P_r$, $S_{melt}$, $\Delta S$, and $E$) during the

2002-2016 period in all sub-basins (GZ to TZL). The colors indicate the degree of correlation with darker

colors reflecting stronger correlations. The dots represent significant correlations ($p \leq 0.05$).

The delayed effect of $P$ on $R$ was investigated considering the entire study period from

2002 to 2016 (Fig. 6). When $\tau = 0$, indicating $R$ responses to $P$ in the same month, the upper

and middle reaches of the basin exhibited higher correlations, with the headwater catchments

of GZ (0.85), ZS (0.84), and WLX (0.74) showing significant correlations. Conversely, LN

(0.67), LG (0.60), and TZL (0.2) had relatively lower correlation coefficient values. A delay of

$\tau = 1$ resulted in better $P$-$R$ correlations for many basins, including all of the nested (non-



headwater) basins DF (0.84), YJ (0.83), LN (0.82), and TZL (0.5) of which the latter two

additionally contain hydropower dams. However, as the lag time increased to 2 and 3 months,

correlations significantly started to weaken, except for the downstream-most TZL, which

showed the most significant correlation (0.61) at a lag of two months. These results demonstrate

that upstream mountainous headwater catchments on average exhibited relatively prompt $R$

responses to $P$, despite seasonal snow storage, whereas $(\tau \geq 1)$ in downstream nested

catchments including those containing hydropower dams that may have effectively altered the

natural precipitation-runoff response.

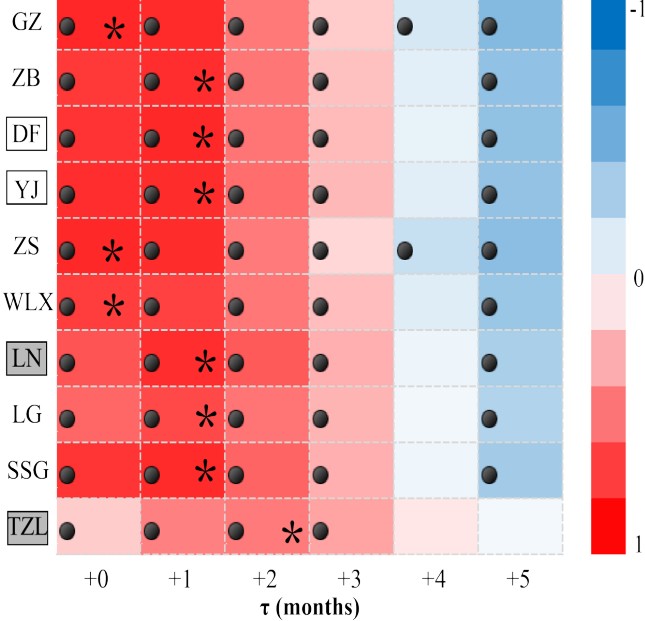

**Figure 6.** Correlation matrix between $R$ and $P$ during the period 2002-2016 in all sub-basins (GZ to TZL;

nested sub-basins have rectangles around their names, with additional grey shading if they contain

hydropower dams). The lagged response in months between $R$ and $P$ is denoted "$\tau$". $\tau$ degree of correlation

with darker colors reflecting stronger correlations. The dots represent significant correlations ($p \leq 0.05$). Each





basin's best fitted $\tau$ is indicated by an asterisk.

### 3.3. Contribution of different factors to runoff variability

Using the Budyko-based variance decomposition method, the influence of various factors

on the intra-annual variance of runoff ($\sigma_R^2$) in the ten sub-basins during the period 2002 to

2016 was quantified, as illustrated in Figure 7. The determination coefficients $R^2$ of all sub-

basins were greater than or equal to 0.9, except for LN (0.83), ZS (0.89), and TZL (0.63). The

slope of the LG, ZB, and DF were 1.0, while SSG and YJ reached a maximum $R^2$ of 0.99.

Substantially, the variance and covariance of $P_r$, $S_{melt}$, $E_0$, and $\Delta S'$ effectively captured $\sigma_R^2$,

which emphasized the significant contribution of these factors to $\sigma_R^2$.

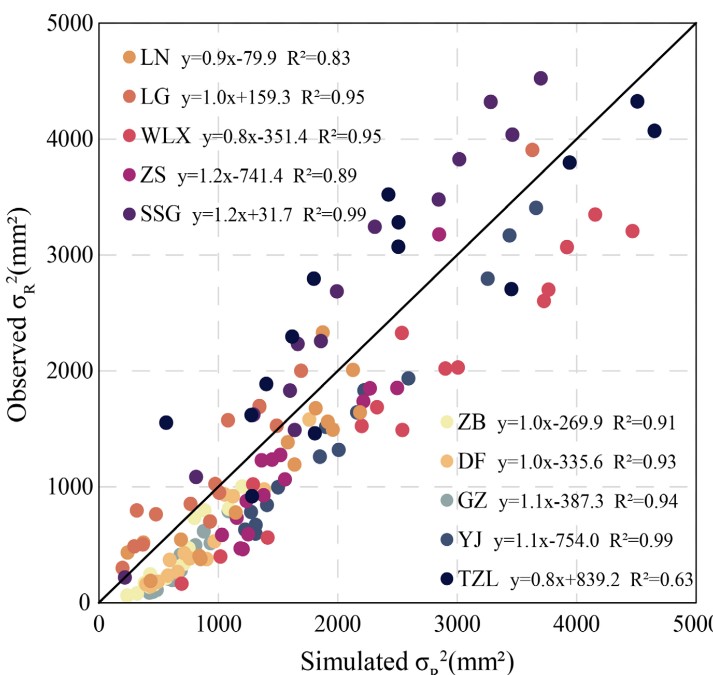

**Figure 7.** Relationships between the observed $\sigma_R^2$ and the simulated $\sigma_R^2$ (Eq. 15) using Budyko-based

framework in all sub-basins.





Figure 8 illustrates the relative contribution matrix of how variances and covariances of water balance terms contribute to the runoff variance $\sigma_R{}^2$. On average, $\sigma_{P_r}{}^2$ was the primary factor driving the intra-annual variance of runoff $\sigma_R{}^2$ in the 10 sub-basins, contributing more than 40%, except for TZL (30.2%) and LG (36.6%). The contribution of $\sigma_{\Delta S'}{}^2$ to $\sigma_R{}^2$ did not exceed 20%, except for TZL (24.3%). Among the covariance contribution of driving factors, $cov(P_r,\Delta S')$ had a significant impact on $\sigma_R{}^2$, reaching a maximum value of 43.3% in LG. The contributions of other factors were limited, within 10%. These findings indicated that variation in $P_r$ and $\Delta S'$ were the primary drivers of intra-annual $R$ variability in the Yalong River basin, with the contribution of $P_r$ variation being most prominent in the middle reaches. The impact of $\Delta S'$ variability was primarily observed downstream, while $S_{melt}$ variability primarily affected the upstream reaches. The magnitude of contribution of the variances and covariances of the water balance terms to $R$ variability is presented in Figure S2. The impact of $\sigma_{E_0}{}^2$ and $cov(P_r,S_{melt})$, on $\sigma_R{}^2$ in the upper and middle reaches of the basin was highly volatile. The contribution of $cov(P_r,\Delta S')$ had a significant negative impact on $\sigma_R{}^2$, reaching a maximum value of -48% in LG, and the contributions of $cov(P_r,S_{melt})$, $cov(P_r,E_0)$, and $cov(S_{melt},\Delta S')$, were all in the range of 5 to -20%.





**Figure 8.** Relative contribution matrix of how variances in different water balance terms ($\sigma_{P_r}^2$, $\sigma_{S_{melt}}^2$, $\sigma_{E_0}^2$, and $\sigma_{\Delta S'}^2$) and the covariances $cov(P_r, S_{melt})$, $cov(P_r, \Delta S')$, $cov(P_r, E_0)$, $cov(S_{melt}, \Delta S')$, $cov(S_{melt}, E_0)$, and $cov(E_0, \Delta S')$ contribute to runoff variance ($\sigma_R^2$) during the period 2002-2016 in all sub-basins. The colors indicate the degree of contribution.

## 4. Discussion

### 4.1. Conributions of $S_{melt}$ and $\Delta S$ to monthly runoff

This study considered the individual contributions of snow storage $\Delta S_{snow}$ and other storage components $\Delta S'$ (reservoirs, lakes, soil water, groundwater) on monthly runoff in the context of the Budyko framework (Bai et al., 2018; Hwang and Devineni, 2022; Shi and Gao, 2022). The findings suggest that $\Delta S_{snow}$ and the associated meltwater term $S_{melt}$ played a prominent role as a source of runoff in the spring hydrological processes, which is consistent with other research findings e.g. by Huang et al. (2018). Present results supported previous results also regarding the dominance of $\Delta S'$ as a source of $R$ in the downstream regions (Huang





et al., 2021; Xu et al., 2012). This dominance has in recent years has been reinforced by increased storage in hydropower reservoirs within the Yalong River basin (e.g., Huang et al.,

2021; Ning et al., 2024; Wu et al., 2024; Xu et al., 2012) However, in contrast to the decreased runoff seasonality that e.g. is found in snow-dominated rivers of central Europe (Rottler et al., 2020), the present study showed that the snow thinning (decreasing $\Delta S_{snow}$ ; Wu et al., 2024) seen in the upper snow dominated sub-catchments of the Yalong River basin is not yet clearly mirrored in runoff seasonality trends (Figure 3). A contributing factor was found to be the

relatively high runoff sensitivity to unfrozen precipitation (rain; Figure 8). Nevertheless, results also showed that, in the upstream Yalong sub-catchments, the seasonal storage of snow still constitutes a large part of the total seasonal water storage (Figure 3). This implies that future runoff seasonality is at risk of decreasing if the climate-driven snow thinning of China's cold regions (Wu et al., 2024; Yang et al., 2015) will continue. Additionally, based on the applied

nested catchment/ Budyko approach, this study contributed to disentangling some of the knowledge gaps related to the acknowledged (Fang et al., 2019) complex spatio-temporal runoff patterns in China's cold regions. In particular, we showed that the snow influence propagates to downstream snow-free regions; as e.g. seen in Figure 9, snowmelt contributes approximately 6% to the annual runoff at the Yalong River outlet (Qi et al., 2022). Furthermore,

the construction of large reservoirs in 1999 and 2013, mainly in the downstream part of the Yalong River basin, has redistributed the intra-annual runoff pattern of the lower basin (Liu et al., 2019). We here showed that reservoirs contribute approximately 7% to the annual runoff at the Yalong River outlet (Figure 9), primarily during January to April. Hence, snow storage and reservoir storage is currently of about the same importance for monthly runoff characteristics





of the entire Yalong River basin, with the relative impact of snow processes and $S_{melt}$ being

considerably larger in its upstream parts. Additionally, since changed snow storage did not yet

decrease runoff variability in the snow-covered upstream part of the basin, it cannot have

contributed to the recent-year decreases total runoff variability of the entire basin either (Figure

9, top panel).

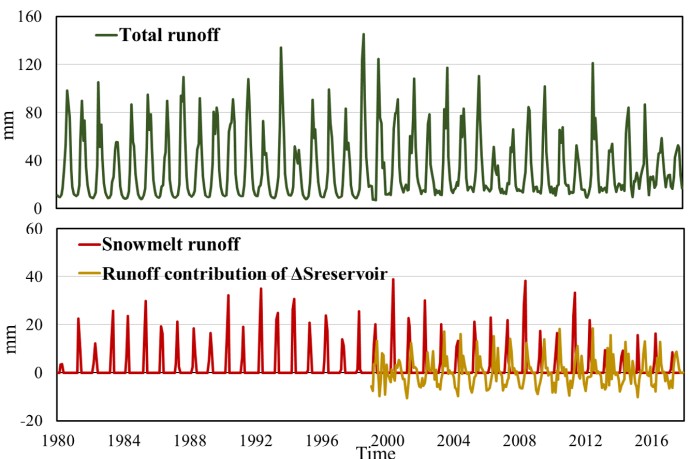


**Figure 9.** Time series of observed total runoff (i.e. monthly discharge volume divided by the total catchment

area), runoff contribution of reservoir water storage change (calculated as monthly reservoir storage volume

divided by total catchment area) and estimated snowmelt runoff (calculated as the monthly snowmelt volume

divided by the total catchment area) from 1980 to 2017 for the whole basin.

Regarding water availability present results showed that, for most of the year, there was

sufficient water for $P_r$ to simultaneously replenish both $R$ and $\Delta S$, resulting in a positive

correlation between $P_r$ and $R$ (Fig. 5a). During the relatively cold dry season, $R$ was then

supplemented by consuming $\Delta S$ (negative correlation in Fig. 5c), thus highlighting the crucial

role of $\Delta S$ in ensuring the availability of water resources throughout the year (Fig. 4). However,

with rising temperatures, $P_r$ and $S_{melt}$ will increasingly be partitioned into $E$ and decreasingly





into $R$ and/or $\Delta S$ during prolonged non-frozen (Condon et al., 2020; Zhang et al., 2015). This disturbs the balance of water supply and demand (Zhang et al., 2016) and calls for further research on the availability of $\Delta S$ to regulate and sufficiently redistribute $R$ among rainy and dry seasons is essential considering the multiple and partially contrasting needs from energy

demand, food demand and other human consumption.

## 4.2. Uncertainties and model performance evaluations

Although the here adopted Choudhury-Yang and Penman-Monteith equations are widely used for actual and potential evapotranspiration, their site-specific applications are associated with uncertainties related to both parameters and processes, such as for instance the under-

studied evaporation mechanism under snow cover conditions (Gan et al., 2022). Since potential errors in $E$-estimations may translate into our $\Delta S'$-results when we close water balances, we here independently check the $E$-estimation consistency by comparing the simulated $E$ from the extended Budyko framework with remote sensing-derived $E$ (GLEAM). The comparison showed good agreement (Table S3), with Nash-Sutcliffe Efficiency (NSE) coefficients

exceeding 0.8 for all 10 sub-basins, indicating high model accuracy and reliability. Furthermore, regarding seasonal water supply and demand, the relationship between the monthly ratio of water demand to water supply ($E/(P_r+S_{melt} - \Delta S')$) and monthly ratio of potential water demand to water supply ($E_0/(P_r+S_{melt} - \Delta S')$) as expressed in the extended Budyko framework was further examined for three representative basins: LN, YJ and ZS (Fig S3).

Water supply constraints typically occurred in the spring, while water demand constraints were more prevalent in the summer and autumn.

Regarding the partitioning of $P$ into corresponding $P_r$ and $P_s$ components including e.g.





snow melt $S_{melt}$ processes, we recognize that there are several challenges of determining

temperature threshold values. These include impacts of solar radiation on snowmelt (Liu et al.,

2017), as well as topography and hydrometeorological factors, particularly in mountainous

regions, where topographical elements such as slope, aspect, and mountain cover exert a

substantial influence on snow melting (Gan et al., 2022). The energy exchange of snow is

furthermore a dynamic process that undergoes temporal variations, leading to discrepancies in

the timescales of degree-day factors across different zones (Zhang et al., 2006). In this study, a

dataset with the spatial distribution of degree-day factors for glaciers in High Mountain Asia

was employed, for which its accuracy had been verified through typical regional simulation

applications (Zhang et al., 2016; Zhang et al., 2017). The fact that the degree-day factor method

yielded satisfactory simulation results at both daily and monthly time scales (Zhang et al., 2016;

2017) underpins the assumption that our can effectively utilized to calculate monthly snow

melt in the here considered Yalong River basin (see also Wu et al., 2024). At the same time, we

acknowledge that remaining, difficult-to-reduce process and parameter uncertainties may have

non-negligible impacts the presented results.

Facing such remaining uncertainties, we independently ascertained that our main (model-

derived) results were consistent with actual, site-specific water storage change outcomes. This

was done by collecting and taking advantage of daily storage and release data from several

reservoirs within the downstream, nested TZL sub-basin (in which $\Delta S_{snow}=0$). This sub-basin

includes the Ertan, Tongzilin and Guandi hydropower reservoirs, which due to their

considerable size together represent a large part of the sub-basin's water storage change $\Delta S$,

making it reasonable to assume that $\Delta S_{Reservoirs} \approx \Delta S$. This hence provides a means to constrain





and verify our estimates regarding the magnitudes and characteristics of $\Delta S$ dynamics, which were derived through water balance closure. We therefore assessed the agreement between the combined monthly storage changes of these reservoirs $\Delta S_{Reservoirs}$ and our estimated storage change $\Delta S$, as illustrated in Figure 10. The figure specifically shows that the estimated dynamics of $\Delta S$ fully encloses the amplitudes and reproduces the trends of observed $\Delta S_{Reservoirs}$.

Regarding the differences in magnitude of storage change, the standard deviation of estimated $\Delta S$ was 86mm, which can be compared with the standard deviation of observed $\Delta S_{Reservoirs}$ of 46mm. This hence suggests that reservoirs account for about 53% of the estimated total water storage changes, with the missing part originating from unmonitored contributions e.g. stemming from storage changes in groundwater reservoirs, unmonitored surface water

reservoirs and soil water. From a methodological viewpoint, we note that the value of 53% also provides an upper limit on our possible overestimation of $\Delta S$, for the unlikely case that storage changes in groundwater reservoirs, unmonitored surface water reservoirs and soil water of TZL would in fact be negligible (i.e. approximately equal to 0), such that the differences in storage changes between the two curves of Figure 10 would be entirely due to errors in snow melt

modelling and water balance closure. However, even for such lowest possible limit value of actual $\Delta S$, the storage volumes are still considerable. These findings hence highlight the critical role that reservoirs, influenced by human activities, can increasingly play in modulating surface water storage in high-mountain areas subject to climate change.



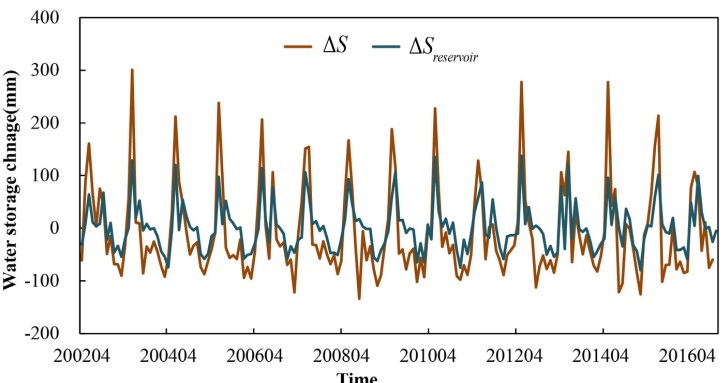

**Figure 10.** Time series of estimated storage change ($\Delta S$) and observed reservoir water storage change ($\Delta S_{Reservoirs}$) from 2002 to 2016 within the nested catchment of TZL.

## 5. Conclusions

This study employed an extended Budyko framework in 10 nested catchments of the cold and mountainous Yalong River basin, to analyze spatio-temporal characteristics in water

balance terms, and to identify main drivers of monthly runoff variability across an elevation gradient (from 5,900 to 1,000 m.a.s.l.). The main findings of this study are:

1) Snow accumulation and snowmelt are main drivers of runoff seasonality in the upper sub-catchments of the Yalong River basin, with propagating impacts also on lower elevation snow-free sub-catchments. These are under increasing additional influence of hydropower

reservoirs, creating a relatively strong altitudinal heterogeneity in drivers of monthly runoff. This is hypothesized to occur also in other world regions including e.g. major European rivers of Alpine origin, although not yet quantified at similarly high spatio-temporal resolution as in the current study.

2) Presently, snow storage and reservoir storage have approximately equal contributions



(6-7% each) to discharge at the Yalong River outlet at its confluence with the Yangtze River, implying that both factors need to be accounted for in predictive models.

3) Snow thinning in the high-elevation, snow dominated sub-catchments of the Yalong River basin is not yet clearly mirrored in time-series of high-elevation runoff seasonality, e.g. due to a considerable runoff sensitivity to unfrozen precipitation.

4) The observed lowered runoff seasonality in the lower Yalong River basin (at its Yangtze River outlet) is therefore not snow-related and hence likely caused by trends in unfrozen precipitation seasonality and/or flow-modulating impacts of constructed reservoirs, natural lakes and groundwater, implying that continued snow thinning may further exacerbate such trends in the future.

5) Regarding lag times, the upstream mountainous headwater catchments of the Yalong basin showed relatively prompt R responses to P (less than a month on average) despite seasonal snow storage, whereas delays were more significant (i.e., more than a month) in downstream nested catchments including those containing man-made reservoirs.

6) Methodologically, we showed by independent verification with reservoir storage data

that the extended monthly Budyko framework could be used to distinguish between water storage and seasonal snow accumulation, which has important implications for understanding dominant runoff processes, and more generally for mitigating adverse effect related to the rapid environmental changes that the Yalong River basin and other cold regions (not least of the Tibetan plateau) are currently experiencing.

## Author Contributions

N.W. and J.J. conceived the idea and designed the research framework. Z.N. carried out


data collection, preprocessing, and method determination. N.W. and H.H. performed data

analysis, graphical visualization, and manuscript preparation. K.Z. and A.N. contributed to

manuscript refinement. All authors have read and agreed to the published version of the

manuscript.

## Acknowledgments

This study was supported by the Fundamental Research Funds for the Central Universities

of China: B240203007 and Fund of National Key Laboratory of Water Disaster Prevention:

524015222.

## Declaration of competing interest

The authors declare that they have no known competing financial interests or personal

relationships that could have appeared to influence the work reported in this paper.

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
