# Peer review of "Altitudinal variation in impacts of snow cover, reservoirs and precipitation seasonality on monthly runoff in Tibetan Plateau catchments"

_Hydrology and Earth System Sciences, 2024_

## Author Comment (AC1)

**Response to RC1:**

**General comments**

The authors of this paper implement an extended Budyko framework for high-elevation

- catchments to understand the changes in water balance characteristics. Additionally, they tried to isolate the major drivers of monthly runoff variability in the region.
  The overall analysis was sound and merits discussion. However, there are significant issues with the general writing of the manuscript, with many convoluted statements.
  My primary concerns are listed below.
- 10 Response: We sincerely thank the reviewer for the valuable and constructive comments. We appreciate the recognition of our work in applying an extended Budyko framework to high-elevation catchments and analyzing the drivers of monthly runoff variability. In response to the reviewer's concerns regarding the quality of writing, we plan to carefully revise the manuscript to improve clarity, rephrase convoluted statements, and
- 15 enhance the overall readability. We will ensure that the scientific content is communicated in a clearer and more concise manner.

A detailed response to each specific comment and a description of the planned revisions are provided below.

**20 Specific Comments**

**Comments 1:** The abstract needs to be more concise. The statements feel overcrowded and convoluted. Please try to split long sentences for better readability.

**Response 1:** We thank the reviewer for the suggestion. In response, we plan to revise the abstract to make it more concise and improve its readability. Specifically, we will

25 split overly long sentences into shorter ones, simplify complex expressions, and restructure the abstract to better highlight the study's objectives, methods, major findings, and significance. The revised abstract will ensure that key messages are communicated more clearly and effectively to the readers.

30 **Comments 2:** How will the phase transition of precipitation (from snow to rain), influenced by a warming climate observed over high-elevation regions, affect runoff characteristics? A brief discussion on this will give more context to the results presented. Also, what about any potential glacier cover in the region?

**Response 2:** We appreciate the reviewer's comment regarding the phase transition of

35 precipitation in high-elevation regions and its potential impact on runoff characteristics. As temperatures rise, precipitation is expected to transition from snow to rain, which makes runoff more sensitive to rainfall (as shown in Figure 8). Moreover, this phase transition could lead to earlier snowmelt, further altering the timing and intensity of runoff. In the revised manuscript, we will discuss the potential effects of climate-driven

40 shifts in precipitation phase (from snow to rain) on runoff characteristics, emphasizing that such transitions could exacerbate the observed decline in runoff seasonality. We acknowledge that these changes in snow storage dynamics could lead to a decrease in runoff seasonality in the future.

Additionally, we have previously analyzed glacier distribution in the study area using glacier data from the Ice and Snow Data Center (as shown in the figure below). The glacier coverage in the region is minimal and can be considered negligible, and thus, we did not consider the impact of glaciers in this study.

Comments 3: Lines 65 to 70: Add references supporting these statements.
 Response 3: We thank the reviewer for pointing this out. We will add relevant references to support the statements made in lines 65 to 70 in the revised manuscript.

**Comments 4:** Line 90: The line should be "A more detailed understanding of intraannual runoff characteristics..."

55

65

reviewer's advice.

**Response 4:** We appreciate the reviewer's comment. We will correct the sentence and revise it to: "A more detailed understanding of intra-annual runoff characteristics..." in the revised manuscript.

60 **Comments 5:** The range of altitudes in the basin has been stated multiple times in the manuscript. However, the source/reference for this information is not mentioned properly.

**Response 5:** We thank the reviewer for pointing this out. We will ensure that the source/reference for the range of altitudes in the basin is properly cited in the revised manuscript.

**Comments 6:** Section 2.1 has many convoluted sentences. Consider simplifying them. For instance, Lines 153 to 157 could be better explained with an equation rather than text. Lines 238 to 242 are not readable.

- **Response 6:** We thank the reviewer for the valuable suggestion. In the revised manuscript, we will simplify the sentences in Section 2.1 to improve clarity and readability. For Lines 153–157, although the runoff calculation process is relatively straightforward, we will rephrase the text to make it more concise and clear, instead of introducing an additional equation. For Lines 238–242, we will rewrite and streamline the text to enhance its readability. We will carefully revise Section 2.1 according to the
  - **Comments 7:** Why is the study period chosen as 2002 to 2016? Were there any

significant changes that happened in the region? Please clarify.

80 **Response 7:** We appreciate the reviewer's comment. We will emphasize the reasons for selecting the study period of 2002 to 2016 in the manuscript. Specifically, we will clarify that this period was chosen because it corresponds to the time during which consistent runoff data were available for all ten sub-basins. Furthermore, this period allows for an evaluation of the impacts of reservoir construction and operation on

hydrological processes, particularly the effects of major reservoirs such as Ertan (1999),
Jinping I (2013), Jinping II (2013), Guandi (2012), and Tongzilin (2015). These developments are crucial for understanding changes in runoff patterns, and this time frame reflects the period before/after the construction of these reservoirs, enabling a thorough assessment of their influence on regional hydrology. We will revise the manuscript to explicitly highlight these reasons.

**Comments 8:** Some of the sub-basins chosen are nested basins. It will be good to mention them and their characteristics in the study area section.

- **Response 8:** We appreciate the reviewer's valuable comment regarding the nested subbasins. As pointed out, this is an important consideration. In the original manuscript, we have already listed the basic characteristics of each sub-basin in Table S2 (in the supplementary material). To provide readers with a clearer understanding of the nested sub-basins, we will move Table S2 from the supplementary material to the main text in the revised manuscript, making it more accessible. This will allow us to present information such as goordinates, basic bydrological, and mateorelogical data in a more
- 100 information such as coordinates, basic hydrological, and meteorological data in a more intuitive manner. Additionally, we will explicitly mention the nested sub-basins and their characteristics in the "Study area and data" section, ensuring that all relevant data is easily available to readers.
- 105 **Comments 9:** Line 145: It is stated that the implementation of the extended Budyko framework is explained in Sect. 2.2. However, Sect. 2.2 in the manuscript is about the study area and data. Also, starting Section 2 with Study Area and Data would be better, and then move to the methodology.

Response 9: We appreciate the reviewer's comments on the structure of Section 2. We initially chose to present the methodology first because it is the key innovation of this study, while the study area was selected as a representative region. However, we understand the reviewer's concern, and in the revised manuscript, we will move the

description of the study area and data to precede the methodology section for clearer presentation of the methodology.

- In the original manuscript, Section 2.1 describes the methodology, while Section 2.2 introduces the study area and data used for implementing the extended Budyko framework. In response to the reviewer's suggestion, we will add the following statement in the revised manuscript: '*In this section, the representative study area and the required data for this study are introduced in Sect. 2.1, while the theoretical*
- 120 framework for attributing runoff variability based on the extended Budyko is described in Sect. 2.2. Table 1 presents the variables and acronyms used in this study'.

Comments 10: Line 248: Check the equation number. Is it 10 or 11?

Response 10: We thank the reviewer for pointing this out. We will carefully check the equation numbering in the revised manuscript, and if any error is found, we will correct it accordingly.

Comments 11: Line 281: The table number should be S2.

**Response 11:** We appreciate the reviewer's careful reading. We will correct the tablenumber in the revised manuscript.

Comments 12: Include elevation information of the outlet stations in Table S2.Response 12: Thank you for the helpful suggestion. We will add the elevation information of the outlet stations to Table in the revised manuscript.

135

**Comments 13:** The labels of the sub-basins are not clear in Figure 2. Consider making them bold/bigger. Also, provide information on the source for the DEM and snow depth data in the figure. Labelling the three figures as (a), (b), and (c) and providing a proper caption for each will bring more clarity.

140 **Response 13:** Thank you for the helpful suggestion. In the revised manuscript, we will improve the clarity of sub-basin labels in Figure 2 by increasing their font size and making them bold. We will also add the data sources for the DEM and snow depth data

directly in the figure or caption. Furthermore, we will label the three sub-figures as (a), (b), and (c), and provide detailed and informative captions for each to enhance readability and clarity.

145

**Comments 14:** Figure 3 indicates consistent negative values for total storage change across all the basins. Explain this.

- **Response 14:** We thank the reviewer for this insightful comment. In our study, we calculated the water balance separately for ten sub-basins. In the upstream sub-basins, which are unaffected by reservoir regulation, the long-term values of  $\Delta S$  fluctuate around zero, indicating that the hydrological system remains in balance under natural conditions. In contrast, the downstream sub-basins affected by reservoir operations (e.g., WLX, LG, TZL) show more pronounced fluctuations in  $\Delta S$ , with a predominance of negative values (i.e., net outflow), which may be attributed to the multi-year regulatory effect of reservoirs in the study area. In addition, this pattern may be partially related to uncertainties in the methodological approach. To address this, we have independently validated the process in Section 4.2 "Uncertainties and model performance evaluations." All results indicate that reservoirs influenced by human activities are
- 160 playing an increasingly important role in regulating surface water storage in alpine regions affected by climate change.

We will clarify this point further in the revised manuscript.

Comments 15: A few new results are introduced in the Discussion section, which should ideally be in the results section. For instance, the total runoff, snowmelt runoff, and the runoff contribution from reservoirs in Figure 9 are not mentioned before. Additionally, Figure 9 shows a time period from 1980 to 2017, which is beyond what was mentioned in the previous results obtained from the study. Please clarify.

Response 15: Thank you for pointing this out. Figure 9 presents a long-term analysis
of the influence of snowmelt and reservoir operations on the intra-annual variability of
runoff at the basin outlet. We included this figure in the Discussion section as a
supplementary analysis to further explain one of our study's key findings—the role of

snowmelt and reservoirs in shaping monthly runoff. Figure 10, on the other hand, was added to independently validate the estimated storage change ( $\Delta S$ ) using reservoir operation data, thereby helping to assess the uncertainty of our water balance results. As both figures are intended to support and reinforce the main findings rather than introduce new directions, we plan to move them to the supplementary materials in the revised manuscript for improved clarity.

Regarding the time period shown in Figure 9 (1980-2017), we acknowledge that it

differs from the primary study period (2002–2016). This extended period was selected specifically to highlight long-term trends in snowmelt reduction and to compare seasonal runoff dynamics before and after reservoir construction. This broader context helps support the interpretation of changes observed during the main study period. We will clearly explain this rationale in the revised figure caption and in the supplementary materials.

200 11100011010.

**Comments 16:** Lines 556-558: A hypothesis on other parts of the world may be made in the discussion part. Or this can be stated as a possible future work.

Response 16: Thank you for your comment. In the discussion section, we have already
made a comparison and hypothesis regarding similar mechanisms in other parts of the world, such as snow-dominated rivers of central Europe. We will revise the manuscript to clarify that this comparison is intended as a potential avenue for future research, and will rephrase the relevant part as: "*Similar elevation-dependent mechanisms may occur in other snow-affected basins, such as Alpine-origin rivers in Europe, though further high-resolution studies are needed to confirm this.*"

---

## Author Comment (AC2)

**Response to RC2:**

**General comments**

This paper presents an evaluation of the monthly water balance using an extended Budyko framework to analyze the contribution of snow storage to runoff seasonality. The topic is timely and offers valuable insights into the understanding of hydrological processes. The manuscript is well written, with a clear description of the methodology and a solid explanation of the results, especially an additional experiment incorporating reservoir data to validate the Budyko approach. The hypotheses employed and limitations of Budyko framework is also discussed. I have only minor comments regarding the paper structure, which I hope will help improve the overall readability of this manuscript.

To my understanding, the idea of including nested catchments is related to identifying the influence of hydropower. If this is correct, I would suggest clarifying this when introducing the nested catchments in the study area section. This will help readers understand the rationale for this design early.

**Response:** We sincerely thank the reviewer for the encouraging and constructive comments. We are glad that you found the study timely and valuable, and we appreciate your recognition of our methodology and presentation.

Regarding your suggestion about the rationale behind using nested catchments, we fully agree that this should be made clearer in the manuscript. As you correctly inferred, the nested catchment design aims to distinguish between natural runoff conditions in the upstream sub-basins and regulated conditions in the downstream sub-basins affected by cascade hydropower operations. This contrast enables a spatially explicit analysis of hydropower impacts within the Budyko framework.

To address this, we will revise Section 2.1.1 (Study Area) to explicitly state the purpose of the nested catchment design and clarify how the upstream and downstream sub-basins differ in terms of both hydrological conditions and anthropogenic influences.

**Specific Comments**

**Comments 1:** Line 29, "increasingly" here better be "increase".

**Response 1:** Thank you for your valuable suggestion regarding the wording in Line 29. To improve clarity and avoid potential misunderstandings, we will revise the sentence as follows:

 "*Results showed that snow accumulation and snowmelt are main drivers of runoff seasonality in the upper sub-catchments, and their effects propagate to the lower-elevation snow-free sub-catchments, which are also subject to additional influence from hydropower reservoirs.*"

We believe this revision better conveys the intended meaning.

**Comments 2:** Line 31, consider rephrasing "other world regions" to "other global regions."

**Response 2:** Thank you for your helpful suggestion. We will revise the phrase "other world regions" to "other global regions" in the manuscript to improve clarity.

**Comments 3:** Line 99, missing a period here at the end of the sentence.

**Response 3:** Thank you for pointing this out. We will add the missing period at the end of the sentence to correct the punctuation.

**Comments 4:** Line 165, Figure 1 needs a more detailed explanation. While it is introduced as an overview in Line 149, the description lacks details on its components (e.g., Part 1, Part 2). I recommend providing a brief explanation of the figure in the text, highlighting how it corresponds to the subsections under 2.1. This would make the structure easier to follow.

**Response 4:** Thank you for your helpful suggestion. We agree that Figure 1 would benefit from a clearer and more detailed explanation in the main text. In the revised manuscript, we will provide a brief but explicit description of Figure 1, highlighting its components (e.g., Part 1, Part 2) and clarifying how it corresponds to the subsections

under Section 2.1. This addition will help improve the logical flow and readability of the methods section.

**Comments 5:** Line 235, regarding the section on cross-correlation, partial correlation is also discussed in the results (Line 355). It would improve consistency to include a brief introduction to partial correlation in the methods section here.

**Response 5:** Thank you for pointing this out. You are correct that partial correlation is discussed in the results section (Line 355), but not sufficiently introduced in the methods section. To improve consistency and clarity, we will revise the manuscript to include a brief introduction to the partial correlation analysis in the methods section. This will help readers better understand its role and relevance in the analysis presented later.

**Comments 6:** Line 239 to 241, the sentences here appear incomplete or unclear. Please revise for clarity and ensure complete sentence structure.

**Response 6:** Thank you for your valuable suggestion. We agree that the original sentences were unclear and lacked complete structure. In the revised manuscript, we will improve the clarity of this section. Specifically, we will revise the text as follows:

" *After undergoing complex surface and subsurface hydrological processes, the catchment responses to precipitation tend to become smoothed and delayed (Brutsaert and Hiyama, 2012). To characterize the hysteresis relationship between monthly precipitation and runoff in the Yalong River basin, we introduced the variable $\tau$, representing the lag time between precipitation events and corresponding runoff responses.*"

This revision enhances the clarity and ensures the sentence structure is complete. Thank you again for your helpful comment.

**Comments 7:** Line 324, Figure 3, there is a purple vertical line at the right border of the figure, which seems unintended. Please check and correct if necessary.

**Response 7:** Thank you for pointing this out. We will correct this issue in the revised manuscript to ensure the figure is clear and accurately presented.

**Comments 8:** Line 355, "downtrending" here seems likely to be "detrending"? As noted in the previous comment, it would be helpful to move this information to the Methods section and include a brief explanation.

**Response 8:** Thank you for your careful reading. We will revise "*downtrending*" to the correct term "*detrending*" in the manuscript. As you suggested, we will also move this information to the Methods section and include a brief explanation of the detrending procedure to enhance clarity and improve consistency across the manuscript.

**Comments 9:** Line 399, here seems a typo error before "degree of correlation."

**Response 9:** Thank you for pointing this out. We intended to express the following: "*The lagged response in months between R and P is denoted "$\tau$". The colors indicate the degree of correlation with darker colors reflecting stronger correlations. The dots represent significant correlations (p≤0.05). Each basin's best fitted $\tau$ is indicated by an asterisk.*"

We will revise this sentence in the manuscript to clarify the intended meaning and eliminate any ambiguity.

**Comments 10:** Line 435, typo error here.

**Response 10:** Thank you for pointing out the typo. We will carefully review and correct the error in the revised manuscript to ensure clarity and accuracy.

**Comments 11:** Line 487, typo error here too.

**Response 11:** Thank you for pointing out the typo. We will carefully review and correct the error in the revised manuscript to ensure clarity and accuracy.

**Comments 12:** Line 571, the phrase "less than a month", does this mean "concurrent" or does it also include a "one-month lag"? Please clarify.

**Response 12:** Thank you for your insightful comment. We agree that the phrase "less than a month" may be ambiguous. To improve clarity, we will revise the sentence as follows:

"*Regarding lag times, the upstream mountainous headwater catchments of the Yalong basin showed relatively prompt runoff (R) responses to precipitation (P), with lag times (τ) of one month or less (i.e., τ ⩽ 1), despite the presence of seasonal snow storage. In contrast, downstream nested catchments, including those containing man-made reservoirs, exhibited more significant delays (i.e., τ > 1).*"

This revision enhances clarity and aligns with your valuable suggestion.

---

## Author Response (AR1)

**Response to RC1:**

**General comments**

The authors of this paper implement an extended Budyko framework for high-elevation catchments to understand the changes in water balance characteristics. Additionally, they tried to isolate the major drivers of monthly runoff variability in the region.

The overall analysis was sound and merits discussion. However, there are significant issues with the general writing of the manuscript, with many convoluted statements. My primary concerns are listed below.

Response: We sincerely thank the reviewer for the valuable and constructive comments, as well as for recognizing our work in applying the extended Budyko framework to high-elevation catchments and investigating the drivers of monthly runoff variability. Regarding the reviewer's concern about the quality of writing, we have carefully revised the manuscript to improve clarity, rephrased convoluted sentences, and enhanced the overall readability. We have ensured that the scientific content is communicated in a clearer and more concise manner.

Below, we provide detailed point-by-point responses to each specific comment, along with descriptions of the corresponding revisions made in the manuscript.

**20 Specific Comments**

**Comments 1:** The abstract needs to be more concise. The statements feel overcrowded and convoluted. Please try to split long sentences for better readability.

**Response 1:** We thank the reviewer for this helpful suggestion regarding the abstract. Following the comment, we have thoroughly revised the abstract to improve conciseness and readability. Specifically, we have split long and complex sentences, simplified phrasing, and ensured clearer expression of the key objectives, methods, and findings.

We hope the revised abstract now provides a clearer and more concise summary of our study and adequately addresses the reviewer's concern.

Comments 2: How will the phase transition of precipitation (from snow to rain), influenced by a warming climate observed over high-elevation regions, affect runoff characteristics? A brief discussion on this will give more context to the results presented. Also, what about any potential glacier cover in the region?

**Response 2:** We thank the reviewer for this important comment regarding the potential effects of precipitation phase transitions and glacier cover on runoff characteristics in high-elevation regions.

As suggested, we have added a brief discussion in the revised manuscript on how the transition of precipitation from snow to rain at high altitudes affects runoff characteristics. Such a shift can increase the sensitivity of runoff to rainfall events (as shown in Fig. 8), and may also lead to earlier snowmelt, thereby altering both the timing and magnitude of runoff. This implies that future runoff seasonality could further decrease if climate-driven snow thinning continues in cold regions of China (Wu et al., 2024; Yang et al., 2015). These points have been incorporated into the discussion section of the revised manuscript (Lines 472-479).

Regarding glacier cover, we analyzed glacier distribution using data from the National Snow and Ice Data Center (as shown in the figure below). Our analysis showed that glacier coverage in the study area is extremely low and negligible. Therefore, glacier melt was not considered as a significant factor in our current study, and we have clarified this in the revised manuscript (Lines 166-168).

We hope these additions adequately address the reviewer's concerns and provide better context for our results.

**Comments 3:** Lines 65 to 70: Add references supporting these statements.

**Response 3:** We thank the reviewer for pointing this out. We have added appropriate references to support the statements in lines 61 to 66 of the revised manuscript.

**Comments 4:** Line 90: The line should be "A more detailed understanding of intraannual runoff characteristics..."

**Response 4:** We thank the reviewer for this correction. We have revised the sentence on line 86 as suggested.

**Comments 5:** The range of altitudes in the basin has been stated multiple times in the manuscript. However, the source/reference for this information is not mentioned properly.

**Response 5:** We thank the reviewer for this helpful comment. To clarify the source of the altitude information, we have added details in lines 198–199 of the revised manuscript, stating that the Digital Elevation Model data at a 1 km resolution were sourced from the U.S. Geological Survey (USGS) (GTOPO30, http://edc.usgs.gov/products/elevation/gtopo30/gtopo30.html). Additionally, we have included a note in the caption of Figure 1 indicating that the elevation data for the 10 sub-basins are based on USGS data.

We hope these revisions adequately address the reviewer's concern.

**Comments 6:** Section 2.1 has many convoluted sentences. Consider simplifying them. For instance, Lines 153 to 157 could be better explained with an equation rather than text. Lines 238 to 242 are not readable.

**Response 6:** We thank the reviewer for the valuable suggestions regarding the clarity of Section 2.1. In the revised manuscript, we have simplified the sentences in the corresponding section (now Section 2.2) to improve clarity and readability.

For lines 153 to 157, given that the runoff calculation process is relatively straightforward, we chose to rephrase the text for greater conciseness and clarity rather than introduce an additional equation. The revised content can be found in lines 208–227 of the revised manuscript.

Regarding lines 238 to 242, we have rewritten and simplified the text to enhance readability. The revised section now reads as follows (Lines 296–308 of the revised manuscript):

"Runoff is influenced by various natural and geographical factors. To further examine the relationships between R and individual hydrological drivers (such as Pr, Smelt,  $\Delta S$ , E), we applied partial correlation analysis, which allows us to control for the effects of other variables. Before performing these correlation analyses, all time series were detrended to minimize the influence of long-term trends that could otherwise bias the estimated relationships (Wu et al., 2024). This approach helps to isolate the independent contribution of each variable to runoff variability, reducing confounding effects from other hydrological processes.

In addition, because runoff responses in the basin tend to become smoothed and delayed after undergoing complex surface and subsurface hydrological processes (Brutsaert and Hiyama, 2012), we introduced the variable  $\tau$  to characterize the lag time between monthly precipitation events and corresponding runoff responses in the Yalong River basin.  $\tau$  was calculated within a range of 0 to 5 months, with statistical significance assessed at a threshold of  $p \leq 0.05$ ."

We have carefully revised Section 2 in accordance with the reviewer's suggestions and hope that these changes have improved the manuscript's clarity.

**Comments 7:** Why is the study period chosen as 2002 to 2016? Were there any significant changes that happened in the region? Please clarify.

**Response 7:** We thank the reviewer for this insightful comment. As clarified in lines 183–188 of the revised manuscript, we selected the study period of 2002 to 2016 because consistent runoff data were available for all ten sub-basins during this time frame.

Additionally, this period was critical for assessing the impacts of reservoir construction and operation on hydrological processes in the region. Major reservoirs, including Ertan Reservoir (1999), Jinping I (2013), Jinping II (2013), Guandi (2012), and Tongzilin (2015), were constructed or began operation during or shortly before this period. These developments significantly altered regional hydrology, making this timeframe crucial for comprehensively evaluating changes in runoff patterns and understanding the hydrological consequences of such infrastructure projects.

We have incorporated this explanation into the revised manuscript and hope it addresses
the reviewer's concern.

**Comments 8:** Some of the sub-basins chosen are nested basins. It will be good to mention them and their characteristics in the study area section.

**Response 8:** We thank the reviewer for this valuable comment regarding the nested sub-basins. As correctly pointed out, this is an important aspect to clarify.

In the original manuscript, we provided the basic characteristics of each sub-basin in Table S2 of the Supplementary Material. To improve clarity and accessibility for readers, we have now moved Table S2 into the main text of the revised manuscript (Table 2). This table includes coordinates, basic hydrological and meteorological data, and information relevant to the nested nature of the sub-basins.

Additionally, we have explicitly mentioned the presence of nested sub-basins and described their characteristics in the "Study Area and Data" section to ensure that readers can easily locate and understand this information.

We hope these revisions address the reviewer's concern.

**Comments 9:** Line 145: It is stated that the implementation of the extended Budyko framework is explained in Sect. 2.2. However, Sect. 2.2 in the manuscript is about the study area and data. Also, starting Section 2 with Study Area and Data would be better, and then move to the methodology.

**Response 9:** We thank the reviewer for this helpful comment regarding the structure of Section 2.

Initially, we organized the methods section first, as the methodological framework represents a key innovation in our study, while the study area was introduced as a representative case. However, we fully understand the reviewer's concern and agree that starting Section 2 with the study area and data would improve the logical flow of the manuscript.

In the revised manuscript, we have moved the description of the study area and data to precede the methodology section. In addition, we have added the following clarifying statement in the introduction of Section 2 to guide readers:

"In this section, the representative study area and the required data for this study are introduced in Sect. 2.1, while the theoretical framework for attributing runoff variability based on the extended Budyko is described in Sect. 2.2. Table 1 presents the variables and acronyms used in this study."

We hope this restructuring and clarification adequately address the reviewer's comment.

**Comments 10:** Line 248: Check the equation number. Is it 10 or 11?

**Response 10:** We thank the reviewer for noticing this detail. We have carefully checked all equation numbers and their references in the revised manuscript and corrected any inconsistencies.

**Comments 11:** Line 281: The table number should be S2.

**Response 11:** We thank the reviewer for pointing this out. As per the reviewer's earlier suggestion, we have moved Table S2 into the main text. Therefore, the reference in line 281 has been updated to Table 2 in the revised manuscript (see line 188).

**Comments 12:** Include elevation information of the outlet stations in Table S2.

**Response 12:** We thank the reviewer for this valuable suggestion. We have added the elevation information of the outlet stations in the third column of Table 2 in the revised manuscript.

Comments 13: The labels of the sub-basins are not clear in Figure 2. Consider making them bold/bigger. Also, provide information on the source for the DEM and snow depth data in the figure. Labelling the three figures as (a), (b), and (c) and providing a proper caption for each will bring more clarity.

**Response 13:** We thank the reviewer for these helpful suggestions. We have revised Figure 2 by improving label clarity, adding data source information, and labeling panels (a), (b), and (c) with updated captions, as shown in the revised figure (Figure 1).

**Figure 1.** (a) Geographic locations, (b) elevation of the 10 sub-basins (based on data from the U.S. Geological Survey), and (c) mean annual snow depth (based on data from the National Tibetan Plateau Data Center) for the period 2002-2016 in the Yalong River basin.

**Comments 14:** Figure 3 indicates consistent negative values for total storage change across all the basins. Explain this.

**Response 14:** We thank the reviewer for this insightful comment. In our study, we calculated the water balance for each of the ten sub-basins separately.

In the upstream sub-basins, which are not significantly affected by reservoir operations, the long-term values of  $\Delta S$  fluctuate around zero, indicating that the hydrological system remains balanced under natural conditions. In contrast, the downstream sub-basins influenced by reservoir operations (e.g., WLX, LG, TZL) show more pronounced fluctuations in  $\Delta S$ , with predominantly negative values. This pattern likely reflects the multi-year regulation effects of reservoirs in the study area (see lines 333–338 of the revised manuscript).

Additionally, this pattern may also partly stem from methodological uncertainties. To address this, we independently validated our approach in Section 4.2, "Uncertainty and Model Performance Evaluations," in lines 537–562.

Overall, our results suggest that reservoirs, as a human intervention, are increasingly playing a critical role in regulating surface water storage in high-mountain regions under the influence of climate change.

Comments 15: A few new results are introduced in the Discussion section, which should ideally be in the results section. For instance, the total runoff, snowmelt runoff, and the runoff contribution from reservoirs in Figure 9 are not mentioned before. Additionally, Figure 9 shows a time period from 1980 to 2017, which is beyond what was mentioned in the previous results obtained from the study. Please clarify.

**Response 15:** We thank the reviewer for raising this important point.

Figure 9 in the original manuscript presented a long-term analysis of the influence of snowmelt and reservoir operations on intra-annual runoff variability at the basin outlet. We included this figure in the Discussion section as a supplementary analysis to further explain one of our study's key findings—the role of snowmelt and reservoirs in influencing monthly runoff.

Similarly, Figure 10 was added to provide an independent validation of our estimated reservoir storage changes ( $\Delta S$ ) using reservoir operation data, thereby contributing to the uncertainty assessment of our water balance results. While these figures were intended to support and strengthen our main findings rather than introduce entirely new research directions, we understand the reviewer's concern regarding the structure of the manuscript. Therefore, in the revised manuscript, we have moved these figures to the Supplementary Material as Figures S3 and S5, respectively, to improve clarity.

Regarding the time period shown in Figure 9 (now Figure S3), we acknowledge that it extends beyond our primary study period of 2002–2016. This longer timeframe was chosen to illustrate the long-term decreasing trend in snowmelt and to compare runoff dynamics before and after major reservoir constructions in 1999 and 2013. We believe this broader context helps to better explain the changes observed during the primary study period. We have clarified this rationale in the revised figure captions and in the Supplementary Material.

We hope these revisions adequately address the reviewer's concerns.

**Comments 16:** Lines 556-558: A hypothesis on other parts of the world may be made in the discussion part. Or this can be stated as a possible future work.

Response 16: We thank the reviewer for this helpful suggestion. In the Discussion section, we have already included a comparison and hypothesis regarding similar mechanisms in other parts of the world, particularly in snow-dominated rivers of Central Europe. To make this clearer and to frame it as a potential avenue for future research, we have revised the manuscript to read: "Similar elevation-dependent mechanisms may occur in other snow-affected basins, such as Alpine-origin rivers in Europe, though further high-resolution studies are needed to confirm this."

We hope this revision adequately addresses the reviewer's comment.

**Response to RC2:**

**General comments**

This paper presents an evaluation of the monthly water balance using an extended Budyko framework to analyze the contribution of snow storage to runoff seasonality. The topic is timely and offers valuable insights into the understanding of hydrological processes. The manuscript is well written, with a clear description of the methodology and a solid explanation of the results, especially an additional experiment incorporating reservoir data to validate the Budyko approach. The hypotheses employed and limitations of Budyko framework is also discussed. I have only minor comments regarding the paper structure, which I hope will help improve the overall readability of this manuscript.

To my understanding, the idea of including nested catchments is related to identifying the influence of hydropower. If this is correct, I would suggest clarifying this when introducing the nested catchments in the study area section. This will help readers understand the rationale for this design early.

**Response:** We sincerely thank the reviewer for the encouraging and constructive comments. We are pleased that you find this study timely and valuable, and we appreciate your recognition of our methodological approach and presentation.

Regarding your suggestion about clarifying the rationale behind using nested catchments, we fully agree that this should be stated more explicitly in the manuscript. As you correctly inferred, the design of nested catchments in our study aims to distinguish the natural runoff conditions in the upstream sub-basins from the regulated conditions in the downstream sub-basins influenced by cascade hydropower operations.

This contrast allows for a spatially explicit analysis of hydropower impacts within the Budyko framework.

To address this, we have revised Section 2.1.1 (Study Area) to explicitly state the purpose of the nested catchment design and to clarify the differences in hydrological conditions and human influences between the upstream and downstream sub-basins.

"The Yalong River is renowned for its abundant hydropower resources. The middle and lower reaches have been designated as a national hydropower base, ranking third among the 13 major hydropower bases in China (Wu and Shen, 2007). In the downstream section, the construction of five major hydropower stations — including Jinping I (2013), Jinping II (2013), Guandi (2012), Ertan (1999), and Tongzilin (2015) — has significantly altered terrestrial water storage and flow regimes (Wu and Shen, 2007).

To assess the spatially varying impacts of hydropower regulation on runoff seasonality, the Yalong River basin was divided into ten sub-basins, forming a nested catchment structure. The upstream sub-basins, located in high-altitude regions, are primarily influenced by seasonal snow accumulation and melt, while glaciers are sparse and thus negligible. These areas experience limited human interference, serving as relatively natural or minimally disturbed reference catchments. In contrast, the downstream sub-basins are heavily regulated by cascade hydropower operations, which have significantly modified water storage dynamics and streamflow patterns. This upstream-downstream contrast enables a comparative analysis of natural versus regulated runoff responses within the Budyko framework. The sub-basins were delineated based on the distribution of hydrological stations (Fig. 1), and the coordinates and basic hydrological and meteorological characteristics of each sub-basin are listed in Table 2."

We hope these revisions improve the clarity and overall readability of the manuscript as suggested.

**Specific Comments**

Comments 1: Line 29, "increasingly" here better be "increase".

**Response 1:** Thank you for your valuable suggestion regarding the wording in Line 29. To improve clarity and avoid potential misunderstandings, we have revised the sentence as follows:

"Results showed that snow accumulation and snowmelt are main drivers of runoff seasonality in the upper sub-catchments, and their effects propagate to the lowerelevation snow-free sub-catchments, which are also subject to additional influence from hydropower reservoirs."

We believe this revision better conveys the intended meaning.

**Comments 2:** Line 31, consider rephrasing "other world regions" to "other global regions."

**Response 2:** Thank you for your helpful suggestion. We have revised the phrase "other world regions" to "other global regions" in the revised manuscript to improve clarity.

**Comments 3:** Line 99, missing a period here at the end of the sentence.

**Response 3:** Thank you for pointing this out. We have added the missing period at the end of the sentence to correct the punctuation.

**Comments 4:** Line 165, Figure 1 needs a more detailed explanation. While it is introduced as an overview in Line 149, the description lacks details on its components (e.g., Part 1, Part 2). I recommend providing a brief explanation of the figure in the text, highlighting how it corresponds to the subsections under 2.1. This would make the structure easier to follow.

**Response 4:** We thank the reviewer for this helpful suggestion. We agree that a clearer explanation of Figure 1 (now Figure 2) and its components was needed to improve the manuscript's readability and to help readers better understand how it corresponds to the subsections under Section 2.1.

Accordingly, we have revised the text to include a more detailed description of Figure 2 (Lines 208-227), explicitly explaining its three main parts and how each part relates to the subsequent sections of the methodology (Sections 2.2.1, 2.2.2, and 2.2.3). This revised description makes the structure of the study clearer and improves the logical connection between the figure and the text.

We hope these changes address the reviewer's comment and enhance the clarity of the manuscript.

**Comments 5:** Line 235, regarding the section on cross-correlation, partial correlation is also discussed in the results (Line 355). It would improve consistency to include a brief introduction to partial correlation in the methods section here.

Response 5: Thank you for this helpful comment. We agree that introducing partial correlation analysis in the methods section improves consistency and clarity. In response, we have revised the manuscript to include a brief explanation of partial correlation analysis and its purpose (Lines 296-308). Specifically, we now clarify that partial correlation was performed to examine the relationships between runoff and individual hydrological drivers while controlling for the influence of other variables, and that all time series were detrended prior to the analyses to avoid biases from long-term trends. We hope this addition enhances the methodological completeness of the manuscript.

**Comments 6:** Line 239 to 241, the sentences here appear incomplete or unclear. Please revise for clarity and ensure complete sentence structure.

**Response 6:** Thank you for pointing this out. In consideration of your comment and other related suggestions, we have reorganized and revised the sentences in this part of the manuscript to improve clarity and ensure complete sentence structure (Lines 296-308). We believe the revised text is now clearer and more precise.

**Comments 7:** Line 324, Figure 3, there is a purple vertical line at the right border of the figure, which seems unintended. Please check and correct if necessary.

Response 7: Thank you for pointing this out. We have corrected this issue in the revised manuscript to ensure the figure is clear and accurately presented.

**Comments 8:** Line 355, "downtrending" here seems likely to be "detrending"? As noted in the previous comment, it would be helpful to move this information to the Methods section and include a brief explanation.

Response 8: Thank you for your careful reading and helpful suggestion. We have corrected the term "downtrending" to "detrending" in the manuscript. In addition, we have moved the explanation of the detrending process to the Methods section and included a brief description to clarify how and why detrending was applied prior to the correlation analyses (Lines 296-308). We believe this improves both the accuracy and clarity of the manuscript.

Comments 9: Line 399, here seems a typo error before "degree of correlation."

**Response 9:** Thank you for pointing this out. We intended to express the following: "The lagged response in months between R and P is denoted " $\tau$ ". The colors indicate the degree of correlation with darker colors reflecting stronger correlations. The dots represent significant correlations ( $p \le 0.05$ ). Each basin's best fitted  $\tau$  is indicated by an asterisk."

We have revised this sentence in the revised manuscript (Lines 420-424) to clarify the intended meaning and eliminate any ambiguity.

Comments 10: Line 435, typo error here.

**Response 10:** Thank you for pointing out the typo. We have carefully reviewed and corrected this error in the revised manuscript to ensure clarity and accuracy (Line 458).

Comments 11: Line 487, typo error here too.

**Response 11:** Thank you for highlighting this typo as well. We have corrected it in the revised manuscript to improve clarity and accuracy (Line 506-507).

Comments 12: Line 571, the phrase "less than a month", does this mean "concurrent" or does it also include a "one-month lag"? Please clarify.

**Response 12:** Thank you for your insightful comment. We agree that the phrase "less than a month" may be ambiguous. To improve clarity, we have revised the sentence as follows:

"Regarding lag times, the upstream mountainous headwater catchments of the Yalong basin showed relatively prompt runoff (R) responses to precipitation (P), with lag times

- ( $\tau$ ) of one month or less (i.e.,  $\tau \leq 1$ ), despite the presence of seasonal snow storage. In contrast, downstream nested catchments, including those containing man-made reservoirs, exhibited more significant delays (i.e.,  $\tau > 1$ )."
- 390 This revision enhances clarity and aligns with your valuable suggestion.